# Bioconjugation of Podophyllotoxin and Nanosystems: Approaches for Boosting Its Biopharmaceutical and Antitumoral Profile

**DOI:** 10.3390/ph18020169

**Published:** 2025-01-26

**Authors:** Carolina Miranda-Vera, Ángela-Patricia Hernández, Pilar García-García, David Díez, Pablo A. García, María Ángeles Castro

**Affiliations:** 1Laboratorio de Química Farmacéutica, Departamento de Ciencias Farmacéuticas, CIETUS, IBSAL, Facultad de Farmacia, Campus Miguel de Unamuno, Universidad de Salamanca, 37007 Salamanca, Spain; cmivedoef@usal.es (C.M.-V.); angytahg@usal.es (Á.-P.H.); pigaga@usal.es (P.G.-G.); pabloagg@usal.es (P.A.G.); 2Departamento de Química Orgánica, Facultad de Ciencias Químicas, Universidad de Salamanca, 37008 Salamanca, Spain; ddm@usal.es

**Keywords:** podophyllotoxin, antitumoral, nanoparticles, polymer-based drug carriers, micelles, liposomes, drug delivery systems

## Abstract

Podophyllotoxin is a natural compound belonging to the lignan family and is well-known for its great antitumor activity. However, it shows several limitations, such as severe side effects and some pharmacokinetics problems, including low water solubility, which hinders its application as an anticancer agent. Over the past few years, antitumor research has been focused on developing nanotechnology-based medicines or nanomedicines which allow researchers to improve the pharmacokinetic properties of anticancer compounds. Following this trend, podophyllotoxin nanoconjugates have been obtained to overcome its biopharmaceutical drawbacks and to enhance its antitumor properties. The objective of this review is to highlight the advances made over the past few years (2017–2023) regarding the inclusion of podophyllotoxin in different nanosystems. Among the huge variety of nanoconjugates of diverse nature, drug delivery systems bearing podophyllotoxin as cytotoxic payload are organic nanoparticles mainly based on polymer carriers, micelles, and liposomes. Along with the description of their pharmacological properties as antitumorals and the advantages compared to the free drug in terms of biocompatibility, solubility, and selectivity, we also provide insight into the synthetic procedures developed to obtain those podophyllotoxin-nanocarriers. Typical procedures in this regard are self-assembly techniques, nanoprecipitations, or ionic gelation methods among others. This comprehensive perspective aims to enlighten the medicinal chemistry community about the tendencies followed in the design of new podophyllotoxin-based drug delivery systems, their features and applications.

## 1. Introduction

Cancer still remains one of the most challenging public health problems worldwide [1,2]. Chemotherapy is the first-choice treatment for this ailment. In spite of great advances, chemotherapy still exhibits huge limitations such as severe side effects or ineffectiveness [3,4]. To overcome these limitations [5], research on nanomedicines or molecular targeted therapy has received considerable attention in the field of oncology. The strategy of molecular targeted therapy refers to the use of drugs or other substances that target specific molecules preventing cancer cells’ growth and proliferation. It also improves the specificity of anticancer agents and reduces non-selective resistance and toxicity [6,7]. Thus, several different targeted therapies have been approved for use in cancer treatment over the past few years [8,9].

The approach to the development of the so-called nanomedicines or nanotechnology-based medicines [10] is based on the inclusion of the targeting drug in a nanoscale delivery system, resulting in an alteration of the physicochemical properties and improving the biopharmaceutical features. Nanoformulation strategies bearing nanomedicines involve the employment of advanced nanocarriers which exhibit enormous ability to deliver therapeutics inside the body, in addition to other advantages such as small size effect, large specific surface area, high reactivity, and quantum effect [10,11,12].

Nanoparticles (NPs) are highly biocompatible and can tolerate a wide variety of cargo and payloads. Furthermore, they can be modified in a manageable manner to enhance drugs therapeutic effectiveness [13], and can be tailored to exhibit various biological properties. These properties make them suitable for use in a wide range of settings, offering safer and more effective methods to deliver chemotherapeutic agents. These days, numerous nanosystems of all kinds of nature (organic, inorganic, or carbon-based) have been described in the literature. These systems have shown the potential to improve biopharmaceutical properties and to enhance specific drug delivery to target cells or organs (Figure 1). To overcome the toxicity associated with conventional chemotherapy, some controlled-release systems that selectively deliver drugs to target tumor cells, are currently under investigation [14].

Furthermore, nanomaterials own a number of advantages as drug carriers: (i) improvement of drug water solubility when dissolved in the bloodstream upgrading the pharmacokinetic and pharmacodynamic properties; (ii) specific delivery of the drug to target tissues or cells, thereby limiting drug accumulation in kidneys, liver, spleen, and other non-targeted organs enhancing therapeutic efficacy; and (iii) potential application as theragnostic agents, combining imaging and therapeutic agents for real-time monitoring the therapeutic efficacy [15,16]. However, although NPs offer many advantages as drug carrier systems, there are still many limitations to be solved such as poor oral bioavailability, inadequate tissue distribution, limited scalability, stability issues during storage, or toxicity [17]. Other properties that may have varied effects on NP pharmacokinetics, clearance, or interactions with other biomolecules, are particle size and surface properties such as charge or hydrophobicity [14]. Regarding the effective NP delivery, there are also extravascular barriers to overcome, as some NPs can extravasate but they do not have the ability to penetrate through the tumor extracellular matrix [15]. Immunogenicity is also a potential hurdle in NP clinical use [14].

On another matter, natural product-derived drugs represent an outstanding source of anticancer compounds [18,19]. Their unique structural characteristics and pharmacological properties [20,21] prompted their consideration as prototypes, leads, or heads of series whose structural modifications have afforded compounds with exceptional pharmacological activities and extraordinary therapeutic possibilities [22].

Among the numerous anticancer natural products described, we have focused our attention on podophyllotoxin, **1**, an aryltetralin lactone belonging to the lignan family that occupies a unique position in the antitumoral natural product’s area (Figure 2) [23]. This compound is isolated from species of the genus *Podophyllum* [24] and shows many attractive pharmacological properties such as antiviral, antioxidant, and immunosuppressive [25]. It has also several uses in dermatology [26]. Nonetheless, the most notorious activity of podophyllotoxin is its antitumoral effect, based on the inhibition of tubulin’s polymerization into microtubules [27,28].

However, in spite of the good antitumor activity of this cyclolignan, it generally shows severe side effects such as bone marrow suppression, hair loss, neurotoxicity, and gastrointestinal toxicity, along with its poor solubility and its lack of selectivity [25].

To overcome these limitations, numerous structural derivatives have been developed. Among them, etoposide (**2**) and its structural analogs teniposide (**3**) and etopophos (**4**) stand out (Figure 2), not only because of their therapeutic potential but also because of the change in their mechanism of action, since they are DNA-topoisomerase II inhibitors instead of tubulin polymerization inhibitors [29,30,31]. Despite being in clinical use, they still have major side effects such as drug resistance and poor bioavailability alongside the generation of specific types of leukemias, caused by chromosomal translocation [29,32]. Hence, further research is necessary to improve their pharmacological properties.

Regarding the relevance of natural products and NPs in cancer treatment, manifold examples of reviews covering the importance of their inclusion in nanosystems can be found in the literature [33,34,35]. As for podophyllotoxin, few reviews highlighting the new trends regarding its bioconjugation have been published over the last few years [27,36,37,38,39]. Nevertheless, no examples have been reported in the literature that thoroughly cover new progress on the inclusion of podophyllotoxin in NPs. Hence, this review is presented with the aim of highlighting the advances accomplished in recent years (2017–2023) regarding the inclusion of podophyllotoxin in drug delivery systems. Although there is a huge variety of nanoconjugates of diverse nature, for those NPs bearing podophyllotoxin as cytotoxic payload, we have considered results concerning organic NPs that were mainly polymer-drug conjugates, micelles, and liposomes. The general advantages and disadvantages of these types of NPs are summarized in Table 1.

In this review, we attempt to describe the latest research on these natural product delivery nanosystems, not only to highlight their antitumor and pharmacokinetic properties but also to provide insights into the synthetic procedures developed to obtain the podophyllotoxin-NPs considered in this review. A schematic representation of the general outline of this review is shown in Figure 3.

## 2. Results

Many podophyllotoxin-prodrugs based on NPs have been developed to increase podophyllotoxin’s aqueous solubility and reduce its side effects. In general, podophyllotoxin is usually attached to the carrier prior to NP’s formation. The synthetic approach of many podophyllotoxin drug delivery systems, mainly polymeric drug carriers and micelles, involves condensation between the C7 hydroxyl group of podophyllotoxin and a carboxylic acid group on the carrier precursors using carbonyldiimidazole (CDI) or ethyl-(*N*′,*N*′-dimethylamino)propylcarbodiimide (EDCI) and 4-(*N*,*N*-dimethylamino)pyridine (DMAP) as coupling reagents, as seen in Figure 1.

As described previously in the Introduction, the main carriers loaded with podophyllotoxin found in literature during the time covered by this review are polymer-based carriers, micelles, and liposomes. The protocols followed to obtain the final NPs once the podophyllotoxin-carrier derivative is synthesized, mostly depend on the chemical and physical properties of the carrier. The different protocols followed to synthesize NP-bearing podophyllotoxin are summarized in Table 2 [46,47,48,49].

### 2.1. Podophyllotoxin Polymer-Based Drug Carriers

Polymer-drug conjugates are organic macromolecules that contain a small bioactive molecule attached to the polymeric backbone through a linker. Depending on the nature of the polymer chain, two groups of polymer conjugates can be distinguished: those with natural polymers such as chitosan or albumin; and those with synthetic polymers like polyethylene glycol (PEG), *N*-(2-hydroxypropyl)methacrylamide (HPMA) and polyglutamic acid (PGA) [42]. The linkers should have special properties in this type of drug delivery system, serving as spacers for molecular co-delivery and/or having specific cleavage properties in response to environmental stimuli [43,44]. In spite of improving the pharmacokinetic and pharmacodynamic properties of podophyllotoxin, sometimes polymeric drug carriers could be themselves cytotoxic and need to stabilize the payload from physiological clearance mechanisms [45].

One of the recently synthesized podophyllotoxin-bearing polymeric carriers was named Celludo (**5**, Figure 4). It was synthesized by Roy et al. [50,51] as shown in previous general Figure 1: acetylated carboxymethylcellulose (CMC-Ac) reacted with podophyllotoxin and poly(ethylene glycol)methyl ether (mPEG) and then, **5** was self-assembled into NPs. Celludo NPs raised the dose of podophyllotoxin that could be administered to mice resulting in enhanced efficacy in mice bearing different MDR tumors. It also increased the accumulation of podophyllotoxin by 500-fold in tumors over time compared to free podophyllotoxin [50,51]. Related to Celludo’s pharmacokinetics, the study revealed that the NP displayed extended blood circulation compared to free podophyllotoxin, with an 18-fold prolonged half-life (12 h of half-life for Celludo versus 0.7 h for free podophyllotoxin). Podophyllotoxin renal clearance was also affected when loaded in the NP. The compound’s clearance was reduced 1000-fold when in Celludo NPs. In terms of biodistribution, in vivo studies in tumor-bearing BALB/c mice demonstrated that tumor uptake of the NPs lasted for 96 h, contrary to free podophyllotoxin. Moreover, NP uptake occurred in the early stages, and the concentration of Celludo reached 120 µg/g at 6 h post-injection. However, Celludo could also accumulate in the liver and the spleen achieving the peak at 24–48 h and then, rapidly decline [50,51].

Ou et al. [52] have synthesized PEG-podophyllotoxin **8**, a new H_2_O_2_-responsive nano-prodrug for podophyllotoxin delivery in which they exploit the ability of PEG of increasing the blood circulation time of NPs and the H_2_O_2_-sensitive characteristics of oxalate ester. Podophyllotoxin was joined through an oxalate ester linkage to intermediate **7** formed by acylation of poly(ethylene glycol)monomethacrylate (PEGMA, **6**) with oxalyl chloride. Following the addition of DSPE-PEG(2000), podophyllotoxin-PEG NPs were obtained by self-assembly, as shown in Figure 2. The yield of podophyllotoxin-PEG entrapment into NPs resulted in 92%, whereas the drug entrapment efficiency was 83%.

The drug release profiles of these NPs showed that the oxalate ester linkage of the prodrug reacted with H_2_O_2_, releasing podophyllotoxin. More than 78% of the parent compound was released from the NP at 1 mM H_2_O_2_, compared to the <20% of podophyllotoxin released from the NP in the absence of H_2_O_2_ after 48 h. The cellular uptake of the prodrug was also studied, and the results indicated that the cellular uptake of podophyllotoxin-PEG NP was time-dependent. Podophyllotoxin liberation also generated higher levels of ROS, promoted mainly due to the release of podophyllotoxin from the NP. Evaluation of the cytotoxic activity of the NPs was also carried out against the CT26 (murine colorectal carcinoma) cancer cell line and the NIH 3T3 (mouse fibroblast) healthy cell line, revealing that after 48 h of incubation, podophyllotoxin-PEG showed lower cytotoxicity against the cell line NIH 3T3 than toward CT26 cells. Regarding NP biodistribution, it followed a time-dependent pattern when fluorescent podophyllotoxin-PEG NPs were injected into BALB/c mice bearing CT26 tumors. The fluorescence intensity peaked at 24 h but was maintained for 72 h [52].

Chen et al. [53] obtained new podophyllotoxin-chitosan NPs as pH-sensitive carriers for encapsulation and release of podophyllotoxin. The synthetic procedure started with a deacetylation reaction of commercial chitosan in alkalis to obtain deacetylated chitosan (CS, **9**). The formation of podophyllotoxin-loaded chitosan NPs (podophyllotoxin-CS NPs, Figure 3) was followed through an ionic gelation method. The synthesis of the NPs was carried out under mild conditions and implied ionic interactions between the positively charged chitosan molecules and negatively charged tripolyphosphate (TPP, **10**), acting as the cross-linker between chitosan and podophyllotoxin. More than 52% of podophyllotoxin was encapsulated, resulting in NPs containing 9.5% podophyllotoxin. Hemocompatibility of the NPs was evaluated showing dose-dependent hemolytic activity with a rate of hemolysis below 5%, which is the critical safe hemolytic ratio for biomaterials according to ISO/Tr7406. Release studies were also carried out, exhibiting pH-dependency. At pH = 3.7, the amount of released podophyllotoxin from the podophyllotoxin-CS NPs was near 80% in 30 h, whereas in neutral and basic mediums the release rate was lower. Regarding cellular uptake of the NPs, it was higher than that of the free drug and it was carried out through electrostatic interactions of protonated chitosan with glycocalyx present on the cell membrane, accelerating the NP internalization process. Finally, in vitro anticancer activity results showed that the podophyllotoxin-CS NPs exhibited better anticancer activity than free podophyllotoxin at all the doses tested when the incubation time was extended to 48–72 h. Cell death studies revealed that cell apoptosis was increased in HepG-2 and MCF-7 cell lines to 34% and 29% for podophyllotoxin-CS NPs compared to free podophyllotoxin (apoptotic rates of 31% and 27% for HepG-2 and MCF-7 cell lines, respectively) [53].

Huang and co-workers [54] synthesized etoposide dual-peptide-modified NPs using pegylated poly(lactide-co-glycolide (PLGA-PEG) as the main polymer, modified with a GPC receptor targeting peptide (AG-peptide), and a peptide with self-penetrating ability (TAT), with the objective to improve etoposide (ETO) chemotherapy in drug-resistant small cell lung cancer (SCLC) cells. The synergistic effect with a PIK3CA small interfering RNA (siRNA) was also studied. For this purpose, four types of polymers were prepared: peptide-free PLGA-PEG, single peptide-conjugated PLGA-PEG-AG and PLGA-PEG-TAT, and dual peptide-conjugated PLGA-PEG-A/T. All of them were used to encapsulate either etoposide or siRNA. The etoposide-loaded NPs (ETO@NPs) were prepared by using a single emulsion solvent evaporation method. siRNA-loaded NPs (siRNA@NPs) were obtained by using the double emulsion solvent evaporation method. The encapsulation efficiency of ETO was higher than 60% in all the polymers synthesized, regardless of the presence of peptides. Cellular uptake was significantly different for PLGA-PEG -AG NPs and PLGA-PEG NPs in CD133(+) H69 cells, implying that AG peptide was responsible for the enhancement of cellular uptake. Cytotoxic activity of ETO@NPs and siRNA@NPs was studied in CD133(+) H69 cells as a model of ETO-resistant SCLC cells. Results showed that the IC_50_ values of all the ETO@NPs were lower than the IC_50_ values of free ETO. However, A/T-NPs-ETO (IC_50_: 15 µg/mL) and A/T-NPs-siRNA (IC_50_: 0.169 µg/mL), exhibited lower IC_50_ than ETO (IC_50_: 81.9 µg/mL) and siRNA loaded single-peptide modified AG-NPs and TAT-NPs (IC_50_: 26.4 µg/mL and 25.2 µg/mL, respectively) which suggested that dual-peptide-modified NPs could efficiently promote NP endocytosis into ETO-resistant cells to inhibit cell growth. Co-treatment with a low dose of ETO@NPs and siRNA@NPs caused less than 26% cell survival after 72 h. These results implied that the combined therapy induced synergistic cytotoxicity in drug-resistant cells and could be promising to reduce not only drug resistance but also cancer recurrence probability [54].

Other approaches for obtaining new podophyllotoxin polymer-based drug carriers are performed when podophyllotoxin is joined by an aliphatic linker or a disulfide spacer to different small organic molecules that serve as inducers for nanoconjugate assembly by different methodologies, as shown in Figure 4. As inducers, the building block 9-fluorenylmethoxycarbonyl (**11**) (precipitation by adding a polar lipid DSPE-PEG(2000)) [55], the non-toxic lipophilic anticancer drug 2-hydroxyoleic acid (**12**) [56] and the natural product cannabidiol (**13**) (by solvent displacement) were used [57] (Figure 4). For all of them, quite interesting cytotoxicity results have been obtained in different cell lines of colon (HT-29) and mesothelium (MSTO-211H and Met-5A). In vivo studies of PEGylated NPs showed good pharmacokinetic and biodistribution properties, the plasma concentrations of prodrug NPs being larger than that of the podophyllotoxin solution, demonstrating great renal clearance resistance and better accumulation at the tumor site [55].

Another similar self-assembly strategy has been followed by Ma and collaborators [58] using a dimer of podophyllotoxin synthesized using disulfide and aliphatic linkers with different lengths. This work demonstrated an increment of solubility and cellular uptake in ovarian A2780S (sensible) and A2780T (resistant) cancer cell lines, (17% and 15.5% cellular uptake, respectively, for dimeric SS-podophyllotoxin NPs), compared to free podophyllotoxin (1.5% and 1.6% cellular uptake, respectively). Moreover, the cellular uptake of the dimeric podophyllotoxin NPs was calculated to be close to 83% and drug release was increased in the presence of 1,4-dithiothreitol (DTT), reaching a 97% podophyllotoxin release rate, compared to the release rate in PBS, which was close to 15%. Nevertheless, NPs’ cytotoxicity against A2780S, A2780T, Hela, MCF-7 and A549 cells was weaker than free podophyllotoxin’s, with IC_50_ values lower than 1 µM [58].

Another interesting approach based on the use of PEGylation was addressed by Islam and collaborators [59]. In this research, a new graphene-based nanocomposite was designed (Figure 5). The introduction of PEG molecules was carried out after the previous carboxylation of graphene oxide. Finally, conjugation with podophyllotoxin was performed. The overall podophyllotoxin loading into the graphene-based nanocomposite was around 25%. The aim of the nanoplatform was to explore its inhibitory activity against α-amylase and α-glucosidase. In vitro studies on the conjugation and release of podophyllotoxin gave satisfactory results, especially when a constant release of the drug was observed for up to 48 h of study. The IC_50_ values of the nanosystem were 7 and 5 mg/mL, against α-amylase and α-glucosidase, similar values for those obtained for the parental compound podophyllotoxin, but holding up the advantage of controlled release [59].

Finally, Ha et al. presented PEGylated podophyllotoxin-derived hydrogels [60]. They obtained an aza-prodrug based on the compound 4′-*O*-demethyl-4β-(4″-aminoanilino)-4-deoxypodophyllotoxin designed to be released in the colon environment. The use of PEG increased the solubility and conferred to the podophyllotoxin derivative an interesting amphiphilic profile that was used by these authors to construct a biocompatible supramolecular hydrogel. In the same work, the hydrogel was loaded simultaneously with other antitumoral agents, using α-cyclodextrin as the building block for the hydrogel construction. In this case, 5-fluorouracil was the therapeutical chosen to be combined with podophyllotoxin. To prove the cytotoxic and biopharmaceutical properties of novel hydrogels, several in vitro assays were performed. For example, as chemical azoreductase mimics colon features, digestive enzymes were used to test the release profile of the podophyllotoxin derivative, reaching above 95% at an almost constant rate. In terms of biodistribution capacity, the study certified that the prodrug and its hydrogel had the ability to release podophyllotoxin derivatives in the colon, in a targeted manner since no podophyllotoxin was found in other organs. Viability assays demonstrated a correlation between cytotoxicity and drug concentration and also synergistic effects of both antitumoral drugs included in the nanocomplex [60].

To better illustrate this section, podophyllotoxin polymer-based carriers described above are collected in Table 3, which includes data related to their synthetic protocols and pharmacological properties when available.

### 2.2. Podophyllotoxin Micelles

Other NP-based drug delivery systems are micelles, which are spherical colloidal NPs that possess a core and shell structure composed of a hydrophobic interior and a hydrophilic exterior. PEG is often used as the hydrophilic shell, although sometimes they can include hydrophobic domains, for example, shells formed by polylactic acid (PLA), poly(lactic-co-glycolic acid) (PLGA), polystyrene, or polylactone. The core, consisting of a hydrophobic domain, acts as a reservoir and protects the drug from being dissolved while the hydrophilic shell region helps to stabilize the hydrophobic core and renders the polymers water-soluble, making the particle a suitable candidate for intravenous administration. Micelles have a small size which allows them to be used for gradual drug release as they are able to avoid the immune system of the patient and filtration of endothelial cells in the spleen [15,17,42]. Due to these properties, several podophyllotoxin micelles have been synthesized in the last years, however, this type of NPs presents some challenges related to instability, potential toxicity, cytotoxicity, and chronic inflammation [15].

A polypeptide-based podophyllotoxin conjugate, PLG-γ-mPEG-podophyllotoxin (**14**, Figure 6), was obtained by Zhou et al. [61] through the condensation of the hydroxyl group of podophyllotoxin with carboxylic groups of the copolymer poly(L-glutamic acid)-γ-methoxy poly(ethylene glycol) (PLG-γ-mPEG) as indicated in Figure 1 and then, conjugates self-assembled into NPs in aqueous solution. The podophyllotoxin release profile of PLG-γ-mPEG-podophyllotoxin micelles was studied in PBS at various pH values, showing that podophyllotoxin kept a sustained and relatively slow-release rate in PBS at pH 7.4–7.0 without enzyme, with less than 10% of the podophyllotoxin released even when the time was extended to 72 h. On the contrary, when incubated with trypsin, micelles showed a significantly increased release trend of podophyllotoxin within 5 h. Cellular internalization of these micelles was studied through the incubation of FI-labelled PLG-γ-mPEG-podophyllotoxin with MCF-7/ADR cells, exhibiting a time-dependent cellular accumulation, noticeable in the cytosol and nuclei surrounding, as the fluorescent intensity was higher at 3 h than at 1 h. In vitro studies revealed that PLG-γ-mPEG-PPT conjugate IC_50_ values against MCF-7/ADR and MCF-7 cell lines were higher than IC_50_ values of free podophyllotoxin (12.3 µM and 5.8 µM, respectively, for podophyllotoxin nanocomposite versus 0.12 µM and 0.12 µM, respectively, for free podophyllotoxin). However, in vivo study with MCF-7/ADR xenograft tumor showed that this conjugate significantly improved antitumor efficacy, with a tumor suppression rate (TSR) of 82.5% and minimal toxicity, as its hemolytic rate was under 10%, compared to free podophyllotoxin (TSR of 37.1% and hemolytic rate of 50%) [61].

PLG-γ-mPEG/PB (**17**, Figure 5) is a PLG-γ-mPEG copolymer (**15**) modified with 4-phenylbutanol (**16**) that was used by Dong et al. [62] to obtain podophyllotoxin-PLG-γ-mPEG/PB micelles (podophyllotoxin-PPB, Figure 6). They used benzene rings to obtain a more stable hydrophobic core for a higher podophyllotoxin encapsulation capacity. Micelles were prepared by synthesizing first PLG-γ-mPEG/PB (PPB, **17**), through the reaction of PLG-γ-mPEG (**15**) and 4-phenylbutanol (**16**) (Figure 5).

Then, podophyllotoxin was loaded into the inner core of the micelles by entrapment through hydrophobic interactions favoured by the phenyl content of the copolymer and purified by dialysis method (Figure 6). In vitro drug release profiles of micelles showed that at pH 5.0, about 80% of podophyllotoxin was released which means that micelles were sensitive to pH values. Also, cytotoxicity was evaluated against the A549 cell line exhibiting that those micelles had comparable cytotoxicity against cancer cells than free podophyllotoxin at the same dosage after 48 h of incubation (IC_50_ values for A549 cell line: 9.4 µM and 6.3 µM for podophyllotoxin-PPB micelles and free podophyllotoxin, respectively) [62].

Li and colleagues [63] incorporated 4′-*O*-demethylpodophyllotoxin into a polymer based on hyaluronic acid (HA), a well-known natural glycosaminoglycan abundant in many tissues. HA-prodrug micelles with 4′-*O*-demethylpodophyllotoxin, named HA-CO-O-podophyllotoxin (**18**, Figure 7), were synthesized conjugating hydrophobic 4′-*O*-demethylpodophyllotoxin to hydrophilic hyaluronic acid (HA) backbone through a one-step esterification reaction and self-assembly in aqueous media [63]. Micelles showed pH-sensitivity, since at pH 5 in PBS, the podophyllotoxin release profiles from micelles increased, reaching 80% of drug release. This fact seemed to be due to the creation of more pore channels, caused by ester bond degradation in the micelles at low pH. These pore channels grew when in contact with water. As a result, a faster liberation of 4′-*O*-demethylpodophyllotoxin from the polymeric prodrug micelles was caused. On another note, HA tumor-targeting ability helped to increase podophyllotoxin’s cellular uptake from the NPs. After incubation for 4 h, the uptake efficiency in MCF-7 cells was over 99%. Additionally, cytotoxicity studies revealed that cell viability decreased in MCF-7 and A549 cells incubated with micelles (IC_50_ values of 1.9 and 4.1 µM, respectively; 2.5- and 1.6-fold lower than IC_50_ values of free podophyllotoxin), but not in normal cells HL7702, which suggests that HA-CO-O-podophyllotoxin micelles could selectively kill tumor cells, which might be related to HA receptor-mediated uptake. Furthermore, the micelles showed great in vivo cytotoxicity, causing excellent tumor inhibition in MCF-7 tumor-bearing mice, and showing a tumor inhibition ratio of 85% [63].

A redox/pH double-sensitive and tumor-targeting drug delivery system for podophyllotoxin based on PEG polymer was developed by Li et al. [64]. They synthesized two podophyllotoxin-conjugates that included a disulfide bridge between podophyllotoxin and polymers. They differ in the absence (PEG-SS-podophyllotoxin, **25a**) or presence (Pep-PEG-SS-podophyllotoxin, **26**) of a transferrin receptor (TfR) targeted peptide, named Pep (Cysteine-Histidine-Alanine-Isoleucine-Tyrosine-Proline-Arginine-Histidin). Simple micelles (Pep-SS-NPs and SS-NPs) were assembled using those two conjugates. Also, micelles without disulfide bridge were obtained (CC-NPs) as control. The synthetic route (Figure 7) toward the podophyllotoxin-prodrug starts with the synthesis of Pep-PEG-NH_2_ (**24**), which in turn was obtained by conjugating Pep cysteine to Mal-PEG-NH_2_ (**23**) by thiol-maleimide click reaction. On their part, SS-podophyllotoxin (**22a**) and its non-sensitive redox counterpart CC-podophyllotoxin (**22b**) (used as control) were synthesized from podophyllotoxin and dithiodipropionic anhydride (DTDPA, **20**, obtained from dithiodipropionic acid **19**), and succinic anhydride (**21**), respectively. Then, the condensation reaction between SS-podophyllotoxin (**22a**) or CC-podophyllotoxin (**22b**) with Pep-PEG-NH_2_ (**24**) and mPEG-NH_2_ yielded the corresponding polymers Pep-PEG-SS-podophyllotoxin (**26**), PEG-SS-podophyllotoxin (**25a**) and PEG-CC-podophyllotoxin (**25b**). Podophyllotoxin prodrug micelles, Pep-SS-NPs, SS-NPs, and CC-NPs were self-assembled using the solvent-evaporation method (Figure 7). Drug loading efficiencies were 13.5%, 14.5%, and 14.2% for Pep-SS-NPs, SS-NPs, and CC-NPs, respectively. Concerning in vitro drug release profiles using a dialysis method, it was observed that podophyllotoxin release from Pep-SS-NPs was extremely slow, less than 10%, within 48 h at pH 7.4 under 20 µM GSH conditions (extracellular environment), while 81.7% of podophyllotoxin was released under a higher GSH concentration of 10 mM. Also, podophyllotoxin release increased at pH 5.0 mainly because the thiopropionate linker was also pH-sensitive. As a control, almost no podophyllotoxin was released from CC-NPs under the same conditions. Cellular uptake studies revealed that it was higher for Pep-PEG-SS-podophyllotoxin micelles treated groups than that of PEG-SS-podophyllotoxin group and that cellular uptake was time-dependent. Additionally, in vitro cytotoxicity studies showed that the common cancer MCF-7 and A549 cell lines, with high expression of P-gp, were highly sensitive to the prodrugs and the IC_50_ values of Pep-SS-NPs were lower than those of SS-NPs and this effect seems to be related to the presence of Pep that facilitated the TfR-mediated endocytosis. Also, in vivo antitumor studies demonstrated that Pep-SS-NPs significantly enhanced therapy efficacy. Tumor inhibition rate for Pep-SS-NPs was 69% whereas the free podophyllotoxin tumor inhibition rate was 40%. In terms of safety, the hemolysis activities of all the micelles synthesized were in the range of 0.01–2 mg/mL, demonstrating the good blood compatibility of these micelles [64].

Hou et al. have created new amphiphilic methotrexate-podophyllotoxin prodrugs that could self-assemble into micelles [65]. They synthesized two amphiphilic prodrugs, one of them in which methotrexate was bound to podophyllotoxin derivate **29a** (podophyllotoxin-SS-OH) through a disulfide bridge and another one in which the union between the two drugs was made through an aliphatic chain using derivate **29b** (podophyllotoxin-CC-OH). The synthetic route is shown in Figure 8. Briefly, podophyllotoxin intermediates podophyllotoxin-SS-OH (**29a**) and podophyllotoxin-CC-OH (**29b**) were synthesized by protecting one of the hydroxyl groups of bis(2-hydroxyethyl) disulphide (BHD, **27a**) or 1,6-hexanediol (**27b**) with a TBDMS group, whereas the other hydroxyl group was linked to podophyllotoxin through a carbonate function using triphosgene. Then, deprotection was followed and intermediates **29a-b** were joined to methotrexate (**30**), obtaining the MTX-SS/CC-podophyllotoxin prodrugs (**31a-b**), that self-assembled into nanoaggregates in aqueous solution without additional carriers [65].

Drug release profiles of prodrugs were studied in the presence of disulfide bond reductive agents such as 1,4-dithiothreitol (DTT) showing that almost 90% of podophyllotoxin was quickly released from MTX-SS-podophyllotoxin (**31a**) when treated with DTT, but no release pattern was observed when MTX-CC-podophyllotoxin (**31b**) was treated with DTT. Additionally, cytotoxic studies against alveolar type I cells (AT1), A549 and KB (human epithelial carcinoma cells) tumor cell lines showed that viability of the three cell lines decreased as the prodrug MTX-SS-podophyllotoxin (**31a**) concentration and incubation time increased. However, free podophyllotoxin showed higher cytotoxicity than MTX-SS micelles. Furthermore, treatment of BALB/c mice bearing 4T1 xenograft tumor with prodrug MTX-SS-podophyllotoxin (**31a**) resulted in decreasing tumor volume over time, with a tumor inhibition of almost 80% after 30 days of treatment [65].

Wen et al. [66] have synthesized FA@PPT-PRA@DOX (folic acid-podophyllotoxin-PRA-doxorubicin) micelles with the aim of obtaining a targeted synergistic drug delivery system, as they included two anticancer drugs, podophyllotoxin and doxorubicin (DOX). Podophyllotoxin was attached to an anti-mitotic cell penetrating octapeptide (PRASHANT, abbreviated as PRA) and modified with folic acid (FA) to target cancer cells specifically. They synthesized first a podophyllotoxin-PRA conjugate (**34**). For doing so, a diacid spacer (**32**) containing a disulfide linkage was attached first, through an ester bond, to podophyllotoxin to get derivative **33**. Subsequently, PRA was joined through an amide bond affording conjugate **34**. PRA octapeptide can strongly bind the exchangeable GTP/GDP binding site of tubulin, inhibiting tubulin polymerization, and inducing apoptotic death. Podophyllotoxin-PRA conjugate (**34**) was self-assembled into vesicles and then loaded with DOX by simply mixing both components at room temperature. Finally, FA-grafted PEG was inserted into the vesicles via hydrogen bonding interaction between FA-PEG and podophyllotoxin-PRA@DOX to obtain FA@PPT-PRA@DOX micelles (Figure 9) [66].

Vesicle drug release studies were carried out in the presence of GSH to investigate GSH-triggered vesicle disruption. Results indicated that both DOX and podophyllotoxin were efficiently released in the presence of GSH and the released amount of parental compound was GSH-concentration dependent. Cellular uptake selectivity was studied in HepG2 and HeLa cancer cell lines overexpressing FA receptors, showing an excellent uptake of the FA@PPT-PRA@DOX micelles in such cell lines. Also, in vitro cytotoxic studies against HepG2 cells and normal cells HL7702 were performed using the newly synthesized micelles. FA@PPT-PRA@DOX micelles showed high selectivity against HepG2 tumor cells, whereas no reduction of cell viability was observed in the normal HL7702 cell line. These results point out a considerable decrease in toxicity toward normal cells and micelle specificity against cancer cell lines. Moreover, FA@PPT-PRA@DOX shower better cytotoxicity than free PPT-PRA (**34**) or the mixture of their elements forming the micelles alone, highlighting their potential ability for synergistic combination therapy [66].

On their part, Xiang et al. have synthesized a new nanoconjugate derived from podophyllotoxin and PEG, PEG-Pept-PPT (**35**, Figure 8) [67]. They formed the nanoparticle by introducing a short peptide H_2_N-CIELLQAR-COOH and mPEG500 through a disulfide linker into the C7 position of podophyllotoxin. The main purpose of the nanoparticle was to specifically target tumor vascular endothelial cells, thanks to the peptide’s ability to mimic selectin E. Release of podophyllotoxin would occur after the GSH-responsive break of the conjugate disulfide bond.

Regarding its cytotoxic activity, the conjugate maintained the inhibitory effect against tumor cell lines tested (MCF-7, 7860, K-562, H1975, and HTC-116). Interestingly, a reduction of cytotoxicity was observed toward normal HUVEC cells compared to free podophyllotoxin. This could mean that podophyllotoxin-NPs were stable around these normal endothelial cells and that podophyllotoxin liberation was specific to tumor cells in which there is an overexpression of GSH due to tumor microenvironment conditions. Apoptosis studies showed that nanoconjugate could promote apoptosis in MCF-7 cells by G2/M phase blockade. This situation did not occur in HUVEC cells in which apoptosis and cell cycle arrest rates were significantly lower than in MCF-7 cells (31% apoptosis by PEG-Pept-PPT in MCF-7 vs. 16% apoptosis in HUVEC cells). On another note, in vivo studies in the MCF-7 xenograft mice tumor model revealed that PEG-Pept-PPT could produce 67% tumor volume inhibition (TMI) after 16 days of conjugate **35** injection. An obvious accumulation in tumor tissues was shown for 0.5–24 h, and there was no apparent toxicity in mice since no weight loss and no alteration of hepatic function markers were recorded during the time covered by in vivo studies. Pharmacokinetic studies revealed that conjugate PEG-Pept-PPT distribution half-life was lower than free podophyllotoxin’s, but its clearance half-life was higher than podophyllotoxin’s (0.12 h and 11.4 h, respectively for distribution half-life; and 49.9 h and 11.4 h for clearance half-life) [67].

Zu et al. [68] have prepared novel micelles of deoxypodophyllotoxin (DPT) (**39**). These lyophilized mPEG-PLA-DPT micelles were obtained by the solvent evaporation-film dispersion method with DPT as the encapsulated drug and mPEG-PLA (methoxy polyethylene glycol-poly(D,L)-lactide, **38**) as the amphiphilic copolymer. Copolymer was obtained from mPEG (**36**) and (D,L)-lactide (**37**) as stated in Figure 10. Then, it was self-assembled in the presence of DPT. mPEG-PLA-DPT micelles showed a very reasonable drug-loading capacity of DPT (98%). The drug release profile was higher in acidic environments, which is a characteristic of tumor cells. However, at pH 7.4, 53% of DPT was released from the micelles. Furthermore, they showed enhanced cytotoxicity against HeLa229 cells compared with free DPT, which would be attributed to the higher internalization of mPEG-PLA micelles than that of free DPT, and to the enhanced release rate in the tumor acidic environment. In vitro release studies indicated that DPT could be released from the micelles by two coexisting mechanisms: diffusion and dissolution. In vivo cellular uptake studies revealed that micelles accumulated into tumor cells in a time- and concentration-dependent manner reaching a saturated state at the dose of 500 ng, 12 h after administration. Micelles also presented great tumor-targeting ability, prolonged half-life, and reduced clearance rate in plasma than other nanocarriers [68].

Novel amphiphilic micelles derived from 4′-*O*-demethylepipodophyllotoxin were obtained by Alliot and colleagues [69]. The amphiphiles were compact micellar carriers formed by a polar head (PEG), cytotoxic molecule 4′-*O*-demethylepipodophyllotoxin, and a lipophilic chain (C_18_) as a hydrophobic core. They were obtained following the procedure shown in Figure 11. First, 4′-*O*-demethylepipodophyllotoxin (**40**) was treated in chloroacetonitrile with a substoichiometric amount of sulfuric acid to obtain a chloroacetamide intermediate. Then, a hydrophobic tail was incorporated at the C4′ position through a Mitsunobu reaction using stearyl alcohol (C_18_), to get **41**. Finally, **41** was treated with amine-derived poly(ethylene glycol) PEG-NH_2_ (**42**), to obtain epipodophyllotoxin-based amphiphile (**43**). Subsequently, conjugate **43** was self-assembled in an aqueous medium to synthesize C_18_-epipodophyllotoxin-PEG micelles. Newly synthesized micelles were injected into mice bearing sub-cutaneous MDA-MB-231 xenografts with 1 wt% of a near-infra-red lipophilic fluorescent dye. Then, 24 h post-injection, fluorescence was still detected in the mice’s bodies indicating that micelles were long-circulating. Furthermore, at the same time, an intense fluorescence signal was found on the tumor mass, suggesting that the C_18_-epipodophyllotoxin-PEG micelles could be an efficient nanocarrier system for in vivo tumor-targeting and vectorization of drugs. Fluorescence was also found in the liver and kidneys, suggesting that micelle elimination could follow both hepato-biliary and urinary pathways. Finally, cytotoxicity studies against MDA-MB-321 revealed that the IC_50_ value of the micelles against this tumor cell line was 21 µM [69].

Novel worm-like morphology polymeric micelles that combined ETO (**2**) and cisplatin (CP) derivatives were obtained by Wan et al. (Figure 12) [70]. Firstly, CP derivatives with aliphatic chains of different lengths (n = 4, 6, 8, and 10) (**44**) at the axial positions were synthesized following a previously reported procedure. Then, ETO-CP-loaded POx (etoposide-cisplatinum-(poly(2-oxazoline) micelles were prepared using the thin-film technique, yielding 98% of ETO encapsulation. Drug release profiles of these worm-like NPs were studied using two techniques: dialysis method using PBS with 40% *w*/*w* BSA (bovine serum albumin) and binding method of micellar drugs with serum proteins. In the case of the dialysis method, results showed that for ETO-C_6_CP-POx micelles, C_6_CP was released faster than ETO and that the release of both drugs was slower than in single-drug micelles. Results of the second method also showed that co-loaded drug micelles retained higher amounts of both drugs. In vitro cytotoxicity and cellular uptake studies suggested that both the antitumor activity and the uptake were enhanced when both drugs were co-formulated, compared to single-drug micelles or to the combination of free drugs, since the drug synergy coefficient (CI) was calculated to < 0.5. Furthermore, in vitro studies against H69AR revealed that the cytotoxicity of the prodrugs increased with the length of the aliphatic chain, with the prodrugs being more active than free cisplatin against H69AR tumor cell line (IC_50_ values 2.8 µg/mL for C_4_CP, 0.31 µg/mL for C_6_CP; 0.06 µg/mL for C_8_CP and C_10_CP; and 3.1 µg/mL for free cisplatin). In vivo safety of ETO-C_6_CP-POx micelles was also evaluated in H69AR tumor-bearing nude mice in which renal toxicity appeared to be less in micelle-treated mice compared to those treated with the free CP and ETO combination, in which mild renal toxicity was observed. Studies relating to the pharmacokinetics of the new synthesized prodrugs revealed that micelles considerably increased the plasma half-life of the bearing drugs, compared to the single-drug micelles (t_1/2_: 6.7 h for C_6_CP vs. 5.1 h for cisplatin and 5.3 h for ETO) [70].

Novel pH-responsive and ROS-generating micelles based on podophyllotoxin were obtained by Li and collaborators [71]. In this work, authors used a thioketal linker joined directly to the natural product. Then, micelles were obtained using a nanoprecipitation method, with an overall 20% of podophyllotoxin loading. They also included a small molecule (cucurbitacin B) to promote ROS generation while the nanoparticle reaches the intracellular milieu. Regarding drug release and cellular uptake patterns of the nanoparticles, they are influenced by ROS and pH variations. As for drug release, it reached 83% when the concentration of H_2_O_2_ was 10 mM; in the absence of H_2_O_2_, podophyllotoxin was not liberated from the nanoparticles. In terms of cellular uptake, it was time-dependent and more prone to occur under the weakly acidic conditions of tumor cells. Considering this trend, the authors proposed a charge conversion of the nanosystem. In this case, the NP surface charge can rapidly change to positive in the tumor’s extracellular environment which promotes its internalization. As for their in vitro cytotoxicity, A549/PTX cells treated with the micelles at pH 7.4 maintained high cellular viability, which was significantly reduced when those cells were treated with the micelles at pH 6.8. The authors performed exhaustive work on the characterization of the NP release and its cytotoxic potential, demonstrating that these NPs respond to oxidative stress and pH stimuli and intensify the cytotoxic effect of podophyllotoxin [71].

In the same fashion, other ROS-responsive NPs (PTV-NP) were synthesized by Liang and Zou [72] adding vitamin K3 as a promoter of drug release. They also obtained a dimer of podophyllotoxin using a thioketal linker. These authors used a nanoprecipitation method where vitamin K3 was included in the NP and a biocompatible triblock copolymer, Pluronic F127, was used as a stabilizing agent. Podophyllotoxin NP loading was 40% wt%. Drug release from NP was H_2_O_2_-dependent since 98% of podophyllotoxin was released when H_2_O_2_ concentration was 10 mM, but less than 5% of podophyllotoxin was released when cells were incubated with PBS 10% at pH 7.4. In vitro studies revealed that for the MCF-7 tumor cell line, the cell viability was reduced to 25% 48 h after treatment. Micelles’ IC_50_ value for MCF-7 cell line was 0.6 µg/mL. In vivo assays also showed better results than free podophyllotoxin both in tumor growth and bodyweight variation in MCF-7 tumor-bearing mice, confirming its upstanding bioavailability and non-toxicity [72].

Recently, Kalinova and collaborators [73] synthesized new micelles of podophyllotoxin and juniper leaf extracts (from *Juniperus virginiana* and *Juniperus sabina* var. *balkanensis*, fam. Cupressaceae), using a biocompatible and biodegradable amphiphilic block copolymer mPEG-*b*-PC (methoxy poly(ethylene glycol)-*block*-polycarbonate). The micelles were obtained first by self-assembling mPEG-*b*-PC macromolecules via the nanoprecipitation method. Then, podophyllotoxin and the two different podophyllotoxin-containing juniper extracts were loaded into nanocarriers using an adsorption technique. Drug loading efficiency for each micelle was 99.8%, 99.6%, and 62.3%, respectively. Bioactivity results of nanoconjugates revealed that podophyllotoxin-loaded micelles showed the highest cytotoxic activity, followed by *J. sabina* and *J. virginiana* leaf extract loaded nanocarriers, against all cancer and normal cell lines tested (IC_50_ values: 0.023, 0.017, 0.014, and 0.004 µg/mL for PPT-loaded micelles; 0.61, 0.32, 0.28, and 0.07 µg/mL for JS-loaded micelles; 3.5, 2.3, 1.2, and 0.57 µg/mL for JV-loaded micelles against MDA-MB-231, A-549, MJ, and HaCaT cell lines, respectively). Furthermore, IC_50_ values of loaded micelles were higher or in a similar range to the IC_50_ values of individual components although with better solubilization than the starting substances. As expected, empty micelles did not show any cytotoxic activity [73].

Podophyllotoxin micelles described in this section are collected and their characteristics are compared in Table 4.

### 2.3. Podophyllotoxin-Liposome Systems

Liposomes also stand as drug carrier systems. They are self-assembled colloidal vesicles with a characteristic lipid bilayer membrane composed of amphiphilic phospholipids. As their composition is similar to cell membranes, liposomes are more biocompatible than other synthetic materials. In addition, another plus of this type of drug carrier system is that liposomes exhibit long circulation time in blood. However, they have disadvantages, such as a low rate of drug release or problems with stability, industrial reproducibility, distribution and removal mechanism, and breakage in vivo, since more time to degrade the liposome vehicle is needed to release the drug, which leads to poor efficacy in vivo [15,40,41].

Ling et al. [74] encapsulated podophyllotoxin in phosphatidylcholine-like liposomes in which fatty acid chains were replaced by two podophyllotoxin molecules, thus forming the named di-podophyllotoxin-GPC liposomes. They were synthesized according to Figure 13, using a lipid film hydration method. The final podophyllotoxin loading of the formed liposomes was around 64%. Several studies of this system revealed that di-podophyllotoxin-GPC liposomes were very stable, showing no premature leakage of their payload in aqueous medium. This stability is mainly owed to the defense mechanism of liposomes, disclosing the podophyllotoxin molecules embedded in the bilayer as the dimeric hydrophobic components of the phospholipid, while the hydrophilic phosphorylcholine head groups were attracted to water forming a membrane surface. In terms of drug release, this stability means that less than 18% of podophyllotoxin was released after 36 h from the drug delivery system when incubated in PBS or FBS, where pH was close to neutrality. However, when liposomes were incubated at pH 5.0, more than 97% of podophyllotoxin was released from the nanocomposite after 36 h, which implied the pH-dependency of the liposomes. Di-podophyllotoxin-GPC liposomes also displayed considerable internalization in MCF-7 cells: the increase in cellular uptake and accumulation of podophyllotoxin exhibited similar time-dependent behaviors to free podophyllotoxin. Furthermore, liposomes exhibited dose-dependent cytotoxic efficacy against all tumor cell lines tested, and their IC_50_ values were calculated as 26 µM, 23 µM, and 24 µM against MCF-7, HeLa, and HepG2 cells, respectively. These IC_50_ values were similar to free podophyllotoxin ones (IC_50_: 22 µM, 18 µM, and 17 µM for MCF-7, HeLa, and HepG-2 cell lines, respectively) [74].

Other podophyllotoxin-loaded liposomes have been synthesized by Wang and colleagues [75,76]. They synthesized the amphiphilic conjugate podophyllotoxin-PBA (**51**) by coupling the podophyllotoxin derivate (**49**) and 4-(4-methylpiperazin-1-ylmethyl)benzoic acid (PBA, **50**) via a hydrolyzable amide linkage, that was co-assembled with a tripeptide cationic lipid (CDO, **52**) giving podophyllotoxin-loaded lipid bilayer NPs (PPCNs), following the dried film ultrasonic procedure (Figure 14). The resulting PPCNs possessed a load of podophyllotoxin of almost 60% by quality. These NPs had an environment-response release, unloosing almost 61% of their podophyllotoxin content in an acidic environment in the presence of hydrolases. When cells were incubated at a physiological pH or a low pH without the presence of hydrolases, less podophyllotoxin liberation from nanoparticles was quantified. In vitro and in vivo biological assays indicated that the tumor inhibition ratio was better for PPCNs than that for free podophyllotoxin. Cytotoxicity of NPs was higher in tumor cells than in normal cells, which can be attributed to the fact that pH is lower in tumor cells and the concentration of hydrolases is higher, thus facilitating podophyllotoxin release from NPs into tumor cells compared to normal cells (IC_50_ values of 16 µM for PPNC and 21 µM for podophyllotoxin in H460 lung cancer cells and >32 µM for PPCN and 1.5 µM for podophyllotoxin in HL7702 normal cells). In terms of the biodistribution of the formed complexes, studies showed that uptake of the PPCNs by H460 cells was time-dependent, accumulating in cell nuclei after 4–10 h of incubation and releasing prodrug podophyllotoxin after 6 h. Most of the complexes accumulated in tumor and lung sites and only a slight amount of them was accumulated in the liver [75].

Wang also investigated these cationic lipid bilayer NPs loaded with both podophyllotoxin and the microRNA miR-424, the latter being a potential tumor suppressor for non-small-cell lung cancer that upregulates the expression of protein programmed death-1 (PD-1) and programmed death ligand 1 (PD-L1) (Figure 14). Results suggested that cellular uptake of miR-424 was increased in H460 cells when miR-424 was absorbed by the PPCN and that the miR-424-PPCN complex significantly downregulated PD-L1 production. Thus, this complex, as a delivery system for both components, was more efficacious than either miR-424 or PPCNs alone. In terms of cell viability, it decreased in H460 cells when they were treated with the complex (63% cell viability in H460 cells treated during 48 h with miR-424-PPCN complexes, compared with 80% viability in H460 cells treated only with PPCN complexes for 48 h), which is consistent with previous results obtained by Wang and colleagues. Regarding the pharmacokinetic properties of the miRNA complexes, the amount of podophyllotoxin in blood was found to be higher for a long time in miR-424-PPCN treated mice than in free podophyllotoxin treated mice (about 18 µg/mL at 24 h in the nanocomposite-treated group but almost none in free podophyllotoxin group) [76].

One of the most recent approaches in the design and synthesis of podophyllotoxin-liposome systems was presented by Niu and co-workers [77], in which podophyllotoxin was included in vesicles based on soybean phospholipids called transfersomes (Figure 15). In this case, the formulation is indicated for topical application. This kind of formulation not only pretends to avoid the systemic secondary effects of podophyllotoxin but also to take advantage of its cytotoxic effects demonstrated against human papillomavirus. For this purpose, transfersomes were synthesized using a thin membrane dispersion method including podophyllotoxin and L-cysteine-deoxycholic acid conjugates (LC-DCA, **55**) apart from soybean phospholipids and tween80. In vitro studies demonstrated that this novel nanoformulation improved topical permeability, with the percentage of drug release being more than 50% at 36 h post-incubation. In vivo studies also showed a high biocompatibility and penetration capacity. Results showed that most of the transfersomes accumulated in the stratum corneum 3 h after administration, and only a part of them reached the active epidermis. After 8 h from administration, nanocomposites were distributed along the epidermis and dermis, corroborating that transfersomes could improve epidermal targeting [77].

All the mentioned podophyllotoxin-liposome systems are collected in Table 5 and pharmaceutical properties can be compared easily.

### 2.4. Other Podophyllotoxin Nanosystems

Finally, fluorescent NPs were obtained by Colombo et al. [78], who developed a novel approach based on drug conjugates that spontaneously self-assembled into photoluminescent NPs. They synthesized a new podophyllotoxin NP (Figure 16) in which they combined 4-aminotetraphenylethylene scaffold (TPE, **58**) as the self-assembly inducer, sebacic acid as the linker and podophyllotoxin as the cytotoxic drug. The 4-aminotetraphenylethylene moiety is an aggregation-induced emission luminophore, which has the ability to show negligible or extremely weak emission in dilute solution but high emission when in solid or aggregate state [79].

They first synthesized a 4-aminotetraphenylethylene scaffold through a McMurry coupling reaction between benzophenone (**56**) and 4-aminobenzophenone (**57**), followed by condensation of compound **58** with sebacic acid to obtain intermediate **59**. Finally, podophyllotoxin was joined to intermediate **59** through an ester bond using coupling reagents. The formation of NPs was carried out using the solvent evaporation protocol. Photoluminescence properties of the NP were measured dispersed in water, showing intense blue fluorescence with a quantum yield of 45%.

Cytotoxic studies of these NPs revealed that either podophyllotoxin conjugate **60** or NP could inhibit cell growth of the ovarian cancer cell lines KURAMOCHI, OVCAR, and OVASHO in the micromolar range (IC_50_ values: 2.3, 7.1 and 45.6 µM, respectively, for podophyllotoxin conjugate **60** and 24.3, 8.8, and 83.7 µM, respectively, for podophyllotoxin NP). However, these IC_50_ values did not improve those of the parental compound podophyllotoxin (IC_50_ values 0.001, 0.0001, and 0.0015 µM, respectively) [78].

## 3. Discussion

The bioactive natural product podophyllotoxin still remains a valuable antitumor agent despite its severe side effects and limited oral solubility. Recently, the growing research on NPs and controlled release systems has opened the doors to the development of new approaches with the aim of improving podophyllotoxin’s antitumor activity, while also attempting to reduce its toxicity and enhancing its poor pharmacokinetic properties.

As discussed alongside the review, researchers have focused on the development of organic NPs bearing podophyllotoxin, either conjugated or encapsulated. Drug polymeric systems, micelles, and liposomes are the most preferred types of organic nanocarriers for podophyllotoxin loading, presumably due to their inherent advantageous features such as low cytotoxicity, biocompatibility, and ease of preparation.

On the subject of the protocols employed to obtain podophyllotoxin-loaded NPs, no standard method has been found. The choice of a particular method is largely influenced by the structure of the podophyllotoxin derivative from which the NP was synthesized. To illustrate, self-assembly techniques are widely employed in micelles when the polymer is directly conjugated with podophyllotoxin [51,61,63,65,66,67,69]. Other strategies found for encapsulating podophyllotoxin into NPs are thin-film techniques [70,74], nanoprecipitation protocols [52,71,72,73], or the ionic gelation method [53].

Regarding the efficiency of podophyllotoxin loading onto drug delivery systems, it depends on the NP type under research. Nevertheless, podophyllotoxin encapsulation generally achieved high yields, typically around 80%. For certain drug delivery systems, the yields were even higher than 95% [68,70].

Podophyllotoxin release profiles have also been investigated, demonstrating that the chemical structure of the nanosystem and the linker play a capital importance in the release of podophyllotoxin. For instance, it can be observed that NPs containing disulfide bonds in their structure are more prone to release podophyllotoxin in the presence of some reducing agents such as GSH or DTT. In this context, disulfide bonds have always been used as linkers that join podophyllotoxin to other components forming the nanostructure, to facilitate the liberation of the drug from the carrier [58,64,65,67,71].

Furthermore, since the tumor environment is more acidic than the normal cell environment [80], some pH-sensitive podophyllotoxin NPs have been synthesized. These nanosystems are able to free podophyllotoxin at acid pH values [53,62,63,68,71,74,75,76]. Release of podophyllotoxin from the nanosystems could occur, in some cases, due to the breakage of the hydrogen bonds formed between podophyllotoxin and the carriers as happened in chitosan NPs, in which the breakage is mainly caused by the protonation of chitosan’s amino group in acidic pH environments [53]. In other cases, podophyllotoxin release is due to micelle size increments caused by the breakdown of ester linkages between podophyllotoxin and hyaluronic acid [63]. The ability of various cellular enzymes like hydrolases, digestive enzymes, or trypsin to break chemical bonds has been used as an alternative form to release podophyllotoxin from the nanocarrier [59,60,61,75]. Thus, the acidic environment in which the NP was embedded facilitated enzyme penetration, and so the enzymes were able to cleave the linkage between NP and podophyllotoxin. Moreover, mechanisms such as diffusion and dissolution can also promote drug release from the nanosystem, as in some mPEG-PLA-DPT micelles [68], although these micelles were also pH-sensitive.

Finally, it is worth noting that some researchers have not only explored the advantages that drug delivery systems offer to decrease the numerous drawbacks of the natural product podophyllotoxin, such as poor solubility or high toxicity; but they have also tried to grant this compound new profitable properties with the creation of drug delivery systems that include other organic molecules like hydroxyoleic acid, cannabidiol, or 4-aminotetraphenylene [56,57,78]. These compounds acted as inducers of NP formation. Furthermore, 4-aminotetraphenylene acted as a luminophore when in the aggregate state, forming a theragnostic nanoparticle, which allowed the follow-up of NP when administered into tumor cell lines. On another note, some podophyllotoxin NPs coated with some sort peptide were created, with the aim of directly targeting these NPs toward certain tumor cell lines, as in reference [64], in which the peptide introduced was a transferrin receptor-targeted peptide, or in reference [67], in which the short peptide introduced could mimic selectin E and target tumor endothelial cells. Following this fashion, miRNAs have also been used for direct-targeting purposes [54,76]. Additionally, other drugs such as 5-fluorouracil [60], methotrexate [65] or cisplatin [70] have been introduced in the nanosystems, either forming hybrid compounds with podophyllotoxin or as part of final NPs, to improve podophyllotoxin nanoconjugates properties.

## 4. Conclusions and Future Perspectives

The excellent cytotoxic properties of podophyllotoxin remain to be exploited and it seems that the development of podophyllotoxin-based nanostrategies is on the rise, with the aim of improving and enhancing these properties. Most of the approaches discussed in this review, drug polymeric systems, micelles, and liposomes, improved cytotoxic properties both in in vitro and in vivo assays, compared to free podophyllotoxin. These facts highlight the potential of this natural product and its derivates and lay the groundwork for its further inclusion into nanosystems for targeting tumor cells.

Going forward, podophyllotoxin-loaded nanocarriers offer manifold advantages compared to the parental compound itself, particularly in terms of biocompatibility, solubility, and selectivity. However, despite the many benefits provided by drug delivery systems, further research is still needed in this field. Many of the difficulties occasionally encountered, such as drug release from the NP, or achieving the cytotoxicity levels of the unencapsulated compound need to be addressed, in order to optimize drug delivery from the carriers, as well as to explore the promising perspectives offered by nanosystems co-loaded with more than one bioactive compound. Finally, it is vital to point out that besides the mentioned challenges, it is of paramount importance to further research in vivo models to assess the potential side effects of the newly synthesized podophyllotoxin-loaded nanoparticles. Additionally, comparative studies of these nanocarriers with the drugs currently in use and stability studies are also essential to determine whether drug delivery systems could offer safer alternatives to conventional chemotherapy. With progress on these fronts, the prospects of podophyllotoxin drug delivery systems will be bright not only in the antitumoral field but also in other therapeutic applications.

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
