# Peer review of "Bioconjugation of Podophyllotoxin and Nanosystems: Approaches for Boosting Its Biopharmaceutical and Antitumoral Profile"

_pharmaceuticals, 2025, doi:10.3390/ph18020169_

Round 1
Reviewer 1 Report
Comments and Suggestions for Authors
The authors here aim at presenting a review on the us of Podophyllotoxin conjugated nanomaterials for use as drug deivery systems. My comments are as follows:
1. Schematic rrpresentation must be improved and put as figure 1 instead of chemical structure so that it represents the idea of the review.
2. A more comprehensive idea of self assembly and guiding factors should be provided.
3. The efficiency of drug conjugation should be mentioned in different systems (not much in detail but a few references)
4. An idea of the PK/PD must be given
5. The authors should mention the following references:
https://www.sciencedirect.com/science/article/abs/pii/S1359644610000334
https://pubmed.ncbi.nlm.nih.gov/31183964/
https://www.sciencedirect.com/science/article/abs/pii/S1359644620302567
6. The authors should highlight if there are any potential drawbacks of this system as compared to either the free drug or the encapsulated drug (instead of conjugated)
Author Response
We sincerely appreciate your review on our article and the valuable feedback and suggestions you have provided that really help to improve its quality. Below you can find our responses to the questions you raised in blue.
The authors here aim at presenting a review on the us of Podophyllotoxin conjugated nanomaterials for use as drug deivery systems. My comments are as follows:
- Schematic rrpresentation must be improved and put as figure 1 instead of chemical structure so that it represents the idea of the review.
Thank you for the suggestion that will improve the review. We have incorporated a new figure (Figure 3) that attempts to show the general scope of the review (at the end of the introduction section); we have also modified the order of figures 1 and 2 and the advantages and limitations of nanoparticles have been included. We have kept Figure 2 (former figure 1) because we truly believe that is very important to highlight the structure of podophyllotoxin and its derivatives as they are present in most of the schemes along the paper.
- A more comprehensive idea of self assembly and guiding factors should be provided.
Thank you for the comment, we would like to say that we only try to give an idea of the synthetic procedures towards nanoparticles, found in the literature, loaded with podophyllotoxin, not to delve into the self-assembly processes. Anyway, following this suggestion, we have included a table (Table 2) in the Results section in which different protocols for nanoparticles synthesis, self-assembly protocols included, are summarized. Also, we had already incorporated for every podophyllotoxin-nanoparticle described in this review the protocol followed by each researcher to obtain the nanoparticle.
- The efficiency of drug conjugation should be mentioned in different systems (not much in detail but a few references)
Thank you for the comment, we have incorporated for each podophyllotoxin-delivery system described the value provided by authors for the efficiency of drug conjugation.
- An idea of the PK/PD must be given.
As suggested, we have included PK/PD data in all cases when such data was provided on the original research articles. Additionally, we have included some tables (Tables 3, 4 and 5) at the end of each type of podophyllotoxin-carriers, in the Results section, that summarizes every nanoparticle described in the article and that allows the comparison among them. The tables contain information related to drug loading efficiency, cellular uptake, cytotoxicity and drug release efficiency for each nanoparticle.
- The authors should mention the following references:
https://www.sciencedirect.com/science/article/abs/pii/S1359644610000334
https://pubmed.ncbi.nlm.nih.gov/31183964/
https://www.sciencedirect.com/science/article/abs/pii/S1359644620302567
As suggested by the reviewer, we have included these references in the introduction section as references 43, 44 and 45.
- The authors should highlight if there are any potential drawbacks of this system as compared to either the free drug or the encapsulated drug (instead of conjugated).
As suggested, we have included data referred to the free drug included in the original articles, independently if the drug was encapsulated or conjugated. We have also provided information about the disadvantages of every drug delivery system described in the Results section. Additionally, we have incorporated Table 1 in which we highlight the advantages and disadvantages of the types of organic nanoparticles in which podophyllotoxin was loaded during the time covered by this review.
Reviewer 2 Report
Comments and Suggestions for Authors
1. The manuscript discusses nanocarriers and their benefits but often lacks quantitative data. It is suggested that the authors include quantitative results from studies on drug loading efficiency, release profiles, and comparative IC50 values for free podophyllotoxin and its nanocarriers across cancer cell lines.
2. Introduction/ Paragraph 3: it should define the advantages of nanomedicine before going to targeted therapy. Authors should cite the following article - https://doi.org/10.3390/ijms251910388 https://doi.org/10.1080/10717544.2023.2284684
3. The introduction does not adequately define the scope of the review. Authors can highlight which specific types of nanocarriers will be the focus or any particular challenges in this field.
4. Figure 2 looks simple; authors are advised to provide detailed mechanisms of nanoparticles in biomedical applications. This will increase the impact and attract the readers.
5. The authors did not mention the methods for synthesizing nanocarriers. Authors are advised to mention the specific approaches (e.g., self-assembly, ionic gelation) used to synthesize NPs.
6. The authors have written many nanoconjugates but fail to compare their performance. Authors are advised to include comparative data, such as drug release rates or cytotoxicity values.
- In some figures, the chemical structures and synthetic pathways, are presented without sufficient explanation. Authors are advised to explain it properly.
- Authors are advised to add a table summarizing different podophyllotoxin-based nanocarriers and their properties, mechanisms, and therapeutic advantages.
- The manuscript focuses primarily on the advantages of nanocarriers but does not address challenges. Authors are advised to include limited scalability, stability issues during storage, and potential toxicity of the nanocarriers themselves.
- Authors are advised to include a section comparing nanocarriers to existing treatments in terms of efficacy, safety, and potential for clinical translation.
11. Decision: Major revision.
Author Response
We sincerely appreciate your review on our article and the valuable feedback and suggestions you have provided that really help to improve its quality. Below you can find our responses to the questions you raised in blue.
- The manuscript discusses nanocarriers and their benefits but often lacks quantitative data. It is suggested that the authors include quantitative results from studies on drug loading efficiency, release profiles, and comparative IC50 values for free podophyllotoxin and its nanocarriers across cancer cell lines .
We greatly appreciate the comment. As suggested by reviewer 2, we have added such information for each podophyllotoxin nanosystem included in this review. We have included Tables 3, 4 and 5 at the end of each podophyllotoxin-carrier type, in the Results section, that summarizes every nanoparticle described and that allows the comparation among them. These Tables contain information related to drug loading efficiency, cellular uptake, cytotoxicity and drug release efficiency for each nanoparticle. Comparative IC50 values for free podophyllotoxin and nanoparticle-podophyllotoxin are included for each nanoparticle system when such information is provided in the original research.
- Introduction/ Paragraph 3: it should define the advantages of nanomedicine before going to targeted therapy. Authors should cite the following article - https://doi.org/10.3390/ijms251910388 https://doi.org/10.1080/10717544.2023.2284684
The introduction section has been reorganized and the advantages of nanomedicine together with some drawbacks relating to nanoparticles have been included and the targeted therapy was reduced to a few lines. Suggested articles have been cited as references 12 and 13 in the introduction section.
- The introduction does not adequately define the scope of the review. Authors can highlight which specific types of nanocarriers will be the focus or any particular challenges in this field.
We have followed the suggestion and as we mentioned in the previous comment, we have re-formulated the introduction section, putting more emphasis on nanoparticles. Also, we have emphasized the specific types of nanoparticles that will be present in the review and we have include Table 1 at the end of the introduction section that highlights the scope of the review. Additionally, a new figure, Figure 3, has been included to better illustrate the scope of the review.
- Figure 2 looks simple; authors are advised to provide detailed mechanisms of nanoparticles in biomedical applications. This will increase the impact and attract the readers.
As suggested, Figure 2 has been modified.
- The authors did not mention the methods for synthesizing nanocarriers. Authors are advised to mention the specific approaches (e.g., self-assembly, ionic gelation) used to synthesize NPs.
We have followed the suggestion by reviewer 2 and we have included a table, Table 2, in which different protocols for nanoparticles synthesis are highlighted in the Results section. Also, we have incorporated for every podophyllotoxin-carrier type included in this review the protocol followed by each researcher to obtain the nanoparticle. We have included now several tables (Table 3, 4 and 5) that also highlight the synthetic protocol followed for the preparation of each nanotransporter.
- The authors have written many nanoconjugates but fail to compare their performance. Authors are advised to include comparative data, such as drug release rates or cytotoxicity values.
Thanks for the comment. We have followed the advice and we have included several tables (Tables 3, 4 and 5) at the end of each section of the different podophyllotoxin-carriers in the Results section. Such compilation of data in tables allow the direct comparation among them. The tables contain information related to drug loading efficiency, cellular uptake, cytotoxicity and drug release efficiency for each nanoparticle system. Additionally, comparative IC50 values for free podophyllotoxin and nanosystem are included in the main text for each nanosystem when such information is provided by authors in the original research.
- In some figures, the chemical structures and synthetic pathways, are presented without sufficient explanation. Authors are advised to explain it properly.
We have followed the advice, and we have now included more information in each figure’s caption about the synthetic route illustrated and about the protocol followed to synthesize each nanoparticle. We have also added the reference in which each figure was based.
- Authors are advised to add a table summarizing different podophyllotoxin-based nanocarriers and their properties, mechanisms, and therapeutic advantages.
Thanks for the good suggestion. We have followed the advice and as mentioned above, we have included several tables, one at the end of each podophyllotoxin-carriers type in the Results section that summarizes every nanoparticle described in the article and that allows the comparation among them. The tables contain information related to drug loading efficiency, cellular uptake, cytotoxicity and drug release efficiency for each nanoparticle.
- The manuscript focuses primarily on the advantages of nanocarriers but does not address challenges. Authors are advised to include limited scalability, stability issues during storage, and potential toxicity of the nanocarriers themselves.
As suggested, we have added general drawbacks of nanoparticles in the Introduction section and a table (Table 1) with general characteristics, advantages and disadvantages of the organic nanoparticles chosen to include podophyllotoxin (polymeric-drug systems, micelles and liposomes). We have also mentioned specific disadvantages along the manuscript for each organic nanoparticle described.
- Authors are advised to include a section comparing nanocarriers to existing treatments in terms of efficacy, safety, and potential for clinical translation.
As far as no podophyllotoxin-nanoparticle is in clinical use, and most of the reported pharmaceutical properties proceed from in vitro assays with few in vivo, we think that the comparison suggested by the reviewer seems to be out of the scope of the review, being a challenge for further studies.
Reviewer 3 Report
Comments and Suggestions for Authors
The group of authors tried to review new podophyllotoxin-based drug delivery nanosystems. The manuscript is needed in the area since podophyllotoxin is not target-specific and must be modified pharmaceutically as a first-generation anticancer agent. The manuscript needs to be revised according to the following comments before further consideration:
1- The first two paragraphs of the introduction are too general and apparent facts, which should be deleted (Lines 32-40). Page 2, Lines 47-55 should be shortened.
2- The authors mentioned the improvement of pharmacokinetic (PK) and pharmacodynamic (PD) properties of podophyllotoxin in the manuscript. Please point out precisely how these characteristics have been improved after modification for each case.
3- The caption of the figures is too brief. Please describe the caption in detail to be understandable independently.
4- When the components are figures from other published papers, please provide appropriate references at the cation as well.
5- The conclusion and future perspective is too long. This section does not need references and should briefly focus on the most promising bioconjugates, which lower the side effects and improve targeted delivery.
6- The manuscript needs to be revised for Grammar in the abstract section and paragraphing style throughout the manuscript. The length of paragraphs is not standard.
7- The abstract is not informative enough. Please classify the bioconjugation methods in the abstract.
Author Response
We sincerely appreciate your review on our article and the valuable feedback and suggestions you have provided that really help to improve its quality. Below you can find our responses to the questions you raised in blue.
The group of authors tried to review new podophyllotoxin-based drug delivery nanosystems. The manuscript is needed in the area since podophyllotoxin is not target-specific and must be modified pharmaceutically as a first-generation anticancer agent. The manuscript needs to be revised according to the following comments before further consideration:
Thank you very much for this comment that we really appreciated
- The first two paragraphs of the introduction are too general and apparent facts, which should be deleted (Lines 32-40). Page 2, Lines 47-55 should be shortened.
We have followed the suggestion, and we have rewritten and reorganized the introduction putting the emphasis mainly on nanoparticles.
- The authors mentioned the improvement of pharmacokinetic (PK) and pharmacodynamic (PD) properties of podophyllotoxin in the manuscript. Please point out precisely how these characteristics have been improved after modification for each case.
Thanks for the comment. As suggested, we now provide in the manuscript all the information about pharmacokinetic and pharmacodynamic properties of the nanoparticles provided in the original articles. Also, we have added more information about each podophyllotoxin-nanosystem regarding drug encapsulation efficiency, drug releasement profiles, cellular uptake and cytotoxicity all along the manuscript and also summarized in new tables 3, 4 and 5.
- The caption of the figures is too brief. Please describe the caption in detail to be understandable independently.
Thanks for the suggestion. We have followed the advice, and we have now included more information in each figure’s caption explaining the synthetic route illustrated and the protocol followed to synthesize each nanoparticle.
- When the components are figures from other published papers, please provide appropriate references at the cation as well.
Thanks a lot for the suggestion. All the figures and schemes are elaborated by us. As suggested, we facilitate now this information on figure captions. And we have also added the reference in which each figure of nanoparticle was based
- The conclusion and future perspective is too long. This section does not need references and should briefly focus on the most promising bioconjugates, which lower the side effects and improve targeted delivery.
As suggested, we have reorganized this section, and we have split it up into two different sections: Discussion, in which we have summarized the trends followed by researches relating to the obtention of nanoparticles bearing podophyllotoxin; and Conclusion and future perspectives section in which we give a current overview on the status of this theme and the approaches that should be tackle in the future to improve the use of podophyllotoxin in this field.
- The manuscript needs to be revised for Grammar in the abstract section and paragraphing style throughout the manuscript. The length of paragraphs is not standard.
We have followed reviewer suggestion and grammar have been revised in the abstract section and all along the manuscript.
- The abstract is not informative enough. Please classify the bioconjugation methods in the abstract.
As suggested, we have included in the abstract more information about the type of nanoparticles and the main synthetic protocols to get them.
Round 2
Reviewer 1 Report
Comments and Suggestions for Authors
Authors have made changes mentioned to them in the previous review round. I recommend this for publication.
Reviewer 2 Report
Comments and Suggestions for Authors
The authors have revised the manuscript following the comments. The manuscript may be accepted for publication in its present form.
Reviewer 3 Report
Comments and Suggestions for Authors
The manuscript is improved after revision.